# How Epistemic Curiosity Influences Digital Literacy: Evidence from International Students in China

**DOI:** 10.3390/bs15030286

**Published:** 2025-02-28

**Authors:** Shaojun Ma, Xuan Jin, Xin Li, Hongming Dong, Xuehang Dong, Bowen Tang

**Affiliations:** 1School of International Education, Tianjin University, Tianjin 300072, China; mashaojun0212@tju.edu.cn (S.M.); dhm@tju.edu.cn (H.D.); dongxh@tju.edu.cn (X.D.); tangbowen_barry@tju.edu.cn (B.T.); 2College of Management and Economics, Tianjin University, Tianjin 300072, China

**Keywords:** international students in China, digital literacy, epistemic curiosity, perceived usefulness, perceiving ease of use

## Abstract

Digital literacy is the core competitiveness and necessary ability that international students should cultivate while studying in China in the context of education digitalization, and this paper mainly explores whether epistemic curiosity can affect the digital literacy of international students in China. Based on the Technology Acceptance Model, this paper introduces the variable of epistemic curiosity, uses questionnaire survey method and quantitative tools (SPSS and AMOS software) to construct a model of the cognition–perception–formation mechanism of international students’ digital literacy in China, and obtains the following conclusions: Firstly, both interest- and deprivation-type epistemic curiosity can directly promote the digital literacy of international students in China. Secondly, this paper discusses how interest- and deprivation-type epistemic curiosity can affect digital literacy under the mediating effect of perceived usefulness. Finally, perceived ease of use can also indirectly promote the relationship between epistemic curiosity and digital literacy of international students in China. The contribution of this paper is to highlight the formation mechanism of digital literacy in cross-cultural contexts and to explore how interest- and deprivation-type epistemic curiosity affect the digital literacy of international students in China. To a certain extent, this paper reveals the potential process of international students in China to use digital resources to transform into digital literacy and also provides useful evidence for the further development of attractive digital resources.

## 1. Introduction

The digitalization of education is an important factor in promoting the transformation of local educational structures and the development of high-quality educational services ([13]). The combination of digital technology and education is becoming a new track for the development of education in all fields around the world and a breakthrough point for continuously shaping new advantages in the development of education ([38]; [48]). With the increasing demand for more and more international students to learn Chinese as a second language ([45]), the digital revolution and resource construction in the field of international Chinese education have achieved remarkable results ([11]; [54]; [53]). For example, in the early stages of the COVID-19 pandemic or when ChatGPT (ChatGPT 3.5)was first introduced, digital smart education platforms developed by the Chinese government and enterprises developed rapidly, and digital technologies such as online teaching, blended learning, and artificial intelligence interactive learning completely changed the traditional learning mode of international students in China ([15]; [27]; [43]). However, there are thresholds and conditions for the application of digital technology in education; in addition to the need for countries or universities to build the necessary digital infrastructure for education, the relevant stakeholders in the education of international students in China also face the challenges of educational digital technology at the behavioral and psychological levels ([37]). The cultivation of digital literacy has an obvious impact on the digital transformation of education in the new era, for example, the European Framework for Digital Competence for Educators (EU); the United Nations Educational, Scientific, and Cultural Organization (UNESCO); the ISTE Standards for Educators (ISTE) in the United States; the Teacher Digital Literacy Framework (UK); and the Digital Literacy of Teachers in China ([17]); however, the impact of digital technology on Chinese as a second language learning from the perspective of international students is still considered insufficient ([55]).

In fact, digital literacy is a concept that has emerged with the development of technologies, such as computer network media, digital twins, and human–computer collaborative teaching ([4]), and because its connotation is iteratively updated according to the global scientific and technological revolution and innovation, and changes in educational resources in different periods, scholars’ knowledge, and understanding of it also have been controversial ([36]; [39]; [40]; [44]), but they all agree on the important role of learners in collecting, processing, and adding value to digital information in the digital age, which promotes the improvement of digital innovation capabilities in the field of education ([3]; [6]; [5]). For this paper, digital literacy for international students in China refers to the digital learning competencies cultivated by international students during their second-language learning in China, which is a comprehensive ability that integrates awareness, application, attitude, and behavior and is fundamental and critical to the Sustainable Development Goals ([11]; [16]; [42]). However, unlike university students in their home countries, international students in China not only face academic pressure and digital competency tests when studying in their host countries, but they also face cultural differences and digital adaptation challenges. [35] ([35]) found that cross-language learners interacting with AI bots can improve their language expression and writing skills, and [55] ([55]) found that international students with high digital literacy can improve their grades and academic performance while reducing their perceived stress about technology.

Although there is a growing interest in the study of digital literacy in the context of second-language learning, there is still little evidence on the mechanism of digital literacy formation of international students and less on the influencing factors of cognitive aspects (e.g., epistemic curiosity). Based on this, this study raises the following research questions: What factors influence the digital literacy of international students in China? How does epistemic curiosity affect the process mechanism of digital literacy formation of international students in China? Based on the Technology Acceptance Model, this study constructs a theoretical model of the digital literacy of international students in China with epistemic curiosity to provide a useful reference for improving the learning effect and perceptual experience of international students in their home countries.

Since it is a relatively new area of research to promote digital literacy in education ([4]), the marginal contributions of this study are as follows: On the one hand, this paper extends the antecedent variables of the Technology Acceptance Model and demonstrates the formation mechanism of international students’ digital literacy in China from the perspective of epistemic curiosity (EC). On the other hand, based on the cross-cultural context and the perspective of international students, this paper discusses the relationship between perceived usefulness, perceived ease of use, interest-type epistemic curiosity (I-type EC), and deprivation-type epistemic curiosity (D-type EC) to explain the logical process of international students using digital resources to transform digital literacy.

The remaining parts of this paper are as follows: Section 2 presents the hypothesis and theoretical background, Section 3 presents the research methodology of the paper, Section 4 presents the results of the paper, Section 5 presents the discussion, and Section 6 is the conclusion.

## 2. Literature Review and Hypothesis

### 2.1. Influence of Epistemic Curiosity on International Students’ Digital Literacy in China

The motivation to explore the knowledge of the unknown world and seek a better understanding of the new world is usually classified as curiosity, and learners are usually more focused on solving real-world problems or life dilemmas under the stimulation of curiosity, and it has been shown that epistemic curiosity is a positive emotion, which is considered to have multidimensional cognitive emotions, and is usually divided into I-type EC and D-type EC ([28]); I-type EC refers to the pleasurable experience and learning motivation brought about by new technologies, while D-type EC refers to the negative feelings caused by the gap between known knowledge and information, and this information deprivation will also stimulate learners’ exploratory behavior ([29]). However, epistemic curiosity is currently considered to be under-researched in second-language learning ([34]). Digital literacy is an essential skill for international students to use digital products for language learning in China, and epistemic curiosity has become an important antecedent for the formation of digital literacy for international students in China ([22]; [23]), as explained below.

I-type EC helps international students to develop digital literacy. On the one hand, international students in China will be exposed to digital teaching devices or mobile digital teaching platforms in the course of their learning in China, and I-type EC can influence the learning attitude and behavior of international students in China, such that they will be more active in seeking knowledge related to Chinese and professional courses, actively participate in course assignments and online communication activities issued by teachers, and lay the foundation for improving the digital literacy of international students in China with interest-driven enthusiasm ([10]; [21]). On the other hand, I-type EC also stimulates international students’ autonomy in digital learning, which can encourage international students to continuously explore new areas and new methods to shape digital literacy, such as using the intelligent voice booth of iFLYTEK combined with Chinese Proficiency Grading Standards to practice pronunciation, and using generative artificial intelligence to conduct knowledge quizzes and heuristic Chinese writing training.

D-type EC is also helpful in cultivating the digital literacy of international students in China. When international students in China become aware of digital literacy deficits or deficiencies, they feel the psychological emotion of “digital deprivation”, which stimulates the curiosity, anxiety, and even the desire to explore among students with D-type EC, which pushes them to work harder to cultivate digital literacy and improve their academic performance ([22]; [46]). Moreover, D-type EC may trigger a sense of anxiety and crisis among international students in China, which prompts them to value learning opportunities more and focus more on improving their digital literacy. Therefore, the following hypotheses are proposed:

**Hypothesis** **1a.**
*I-type EC has a positive impact on the digital literacy of international students in China.*


**Hypothesis** **1b.**
*D-type EC has a positive impact on the digital literacy of international students in China.*


### 2.2. The Mediating Role of Perceived Usefulness

Perceived usefulness in this paper refers to the perception of the existence of digital service products in the process of digital learning, which means that when international students in China perceive that digital learning technology, data analysis, and other knowledge can be helpful for learning, they tend to improve their digital literacy ([2]; [1]). Cognition is the antecedent of perception, and this paper argues that perceived usefulness plays a mediating role between epistemic curiosity and the formation of digital literacy among international students in China ([23]). For I-type EC, when international students are interested in digital awareness, data thinking, and application of digital technology, perceived usefulness is an important mediating factor to ensure the continuity and effectiveness of digital learning, which can further promote international students to transform I-type EC into practical action, so that they consciously cultivate digital skills to improve their learning efficiency, enhance their professional competitiveness or improve their quality of life, and further promote the formation of digital literacy of international students in China ([41]). For D-type EC, the experience of “digital deprivation” is exacerbated by the mediating effect of perceived digital technology, and when students believe that they can improve their digital competitiveness or solve real-world problems by compensating for the lack of these digital skills, they will work harder to improve their digital literacy ([9]). On this basis, the following hypotheses are proposed:

**Hypothesis** **2a.**
*Perceived usefulness has a positive mediating effect on the relationship between I-type EC and international students’ digital literacy.*


**Hypothesis** **2b.**
*Perceived usefulness has a positive mediating effect on the relationship between D-type EC and international students’ digital literacy.*


### 2.3. The Mediating Role of Perceived Ease of Use

Perceived ease of use can be defined as the degree to which international students in China are able to master and use digital learning tools with relative ease and comfort. Previous studies have shown that perceived ease of use reflects the usability of digital educational technologies and facilities and is also a direct factor in learners’ willingness to adopt them ([2]; [32]; [51]). Based on previous studies’ findings, we argue that perceived ease of use can be a mediating variable between epistemic curiosity and the formation of digital literacy among international students in China. For I-type EC, if international students find that digital learning tools or platforms are easy to use, intuitive, and understandable, and the cost of learning is relatively low, they are more likely to turn their I-type EC into a continuous investment of time, and the positive experience not only enhances international students’ learning motivation but also promotes their ability to apply digital skills in real life, which in turn helps cultivate the digital literacy of international students in China. For D-type EC, it often causes digital discomfort or anxiety among international students in China due to poor information, which increases the difficulty and resistance of learning to a certain extent, and perceived ease of use is also an important mediating variable, which can help international students in China reduce the psychological burden of learning thresholds, and then overcome learners’ language barriers and technical anxiety, so as to promote their active participation in learning and communication activities for the cultivation of digital literacy, and further improve digital literacy. On this basis, the following hypotheses are proposed:

**Hypothesis** **3a.**
*Perceived ease of use has a positive mediating effect on the relationship between I-type EC and international students’ digital literacy.*


**Hypothesis** **3b.**
*Perceived ease of use has a positive mediating effect on the relationship between D-type EC and international students’ digital literacy.*


## 3. Method

### 3.1. Instrument Development

This paper focuses on the following variables: digital literacy (DL), interest-type epistemic curiosity (I-type EC), deprivation-type epistemic curiosity (D-type EC), perceived usefulness (PU), and perceived ease of use (PEU). Therefore, the questionnaire is mainly composed of three parts; the first part is the digital literacy scale for international students in China, the second part is the perception scale of digital teaching technology for international students in China, which includes the dimensions of EC, PU, and PEU, and the third part is the basic information data of international students in China, including gender, age, study stage, major, cumulative residence time in China, and Chinese study time, which are the control variables of this paper.

Digital literacy (DL): The digital literacy scale in this paper is adapted from the Global Framework of Reference on Digital Literacy Skills for Indicator 4.4.2 published by UNESCO in 2018, which consists of 18 questions in five parts, including digital information and data literacy, digital communication and collaboration, digital content creation, digital security, and digital problem solving ([49]). All questions are measured on a five-point Likert scale; 1 is strongly disagree, 5 is strongly agree, and the higher the score, the deeper the level of DL, such as my ability to quickly, accurately, and comprehensively retrieve the required curriculum resources from the internet or databases; I was able to present my ideas in the form of original documents, audio, and video using multimedia tools (such as video editing software, PPT) and other tools; I do not divulge my own and others’ information or sensitive data on the internet, etc. (see Section A.1).

Epistemic curiosity (EC): Following [30] ([30]), [28] ([28]) and [29] ([29]) this paper divides epistemic curiosity into two subscales with a total of 10 items, according to the two-dimensional structure of interest and deprivation, and the questions are scored using the five-point Likert scale, where 1 indicates strongly disagree, 5 indicates strongly agree, and the higher the score, the higher the degree of EC. I-type EC focuses on exploratory behaviors driven by one’s own curiosity, such as when I learn new technologies, I like to dig deeper and discover more applications of them; I like to learn things like digital courses, tools, or software that I am not familiar with. D-type EC focuses on exploratory behavior driven by negative feelings caused by poor information, such as when I encounter new technical difficulties in the course, I have to solve them before I can take a break or even spend hours on them; I feel frustrated when I cannot solve the digital technology problem in the course, so I work harder to learn, and so on (see Section A.2).

Perceived ease of use (PEU) and perceived usefulness (PU): The Technology Acceptance Model is a common basic theory of digital technology use and behavioral intention perception in language education ([7]; [47]). PEU and PU are the two core variables of this theory. In this paper, the PEU and PU scales are modified from [12] ([12]) and [50] ([50]). The scale has a total of 9 items, and the questions are scored using a five-point Richter scale, where 1 is strongly disagree, 5 is strongly agree, and the higher the score, the higher the respondent’s perception of the digital technology in the course. Examples of the PEU and PU in this section are It is very easy for me to use digital technology to learn what I am interested in; It is very easy for me to use digital technology to improve my learning and productivity; Using digital technology allows me to complete my study tasks more efficiently, etc. (see Section A.3).

Based on the literature review and the above analysis, this paper constructs a framework as shown in Figure 1.

### 3.2. Participants and Procedure

The participants in this paper are mainly international students studying in China. At present, there are three categories of Chinese students studying in China: preparatory students, undergraduate students, and graduate students, among which preparatory students mainly study Chinese language and Chinese culture, and undergraduate and graduate students mainly study professional knowledge, and they study and live together with Chinese students. The research samples of this paper were collected through online questionnaires, and the data were sampled from October to November 2024 using a snowballing and peer-led method, of which the first stage was mainly distributed to professional course teachers, international students’ education management advisors, and free distribution in each university, and the second stage was mainly collected through dormitory management teachers and compatriots of international students in China to ensure the cross-cultural background of international students’ data. A total of 398 questionnaires were collected through the online platform of Questionnaire Star, and a total of 353 valid data samples of international students were collected in this study after excluding invalid questionnaires, such as a too-short completion time (less than 120 s), single option, and repeated completion of the same IP.

In terms of gender, 163 students (46.2%) were male, and 190 students (53.8%) were female; in terms of age, 6 students were under 18 years old (1.7%), 164 students were 18~22 years old (46.5%), 87 students were 22~25 years old (24.6%), 63 students were 25~30 years old (17. 8%), and 33 students were over 30 years old (9.3%); in terms of academic qualifications, there were 5 students (1.4%) in high school or below, 21 students (5.9%) in preparatory departments, 217 undergraduates (61.5%), 78 master’s students (22.1%), and 32 doctoral students (9.1%). In terms of majors, there were 137 students (38.8%) majoring in Chinese, such as Chinese, teaching Chinese to speakers of other languages, and intercultural education, 40 students (11.3%) majoring in other social disciplines (philosophy, economics and management, law, education, and history), and 176 students (49.9%) majoring in science, engineering, agriculture, medicine, etc. In terms of cumulative residence time in China, 108 students (30.6%) have less than 1 year, 148 students (41.9%) have 1–2 years, 58 students (16.4%) have 3–4 years, 26 students (7.4%) have 5–6 years, and 13 students (3.7%) have more than 6 years. In terms of Chinese learning time, 78 students (22.1%) had less than 1 year, 72 students (20.4%) had 1–2 years, 76 students (21.5%) had 3–4 years, 24 students (2.8%) had 5–6 years, and 103 students (29.2%) had more than 6 years.

### 3.3. Data Analysis

In this paper, SPSS 27.0 and AMOS 27.0 software are mainly used for data analysis ([8]; [19]). The overall Cronbach’s of the questionnaire in this paper is 0.963, and the reliability of each dimension scale was 0.892 for I-type EC, 0.797 for D-type EC, 0.839 for PEU, 0.900 for PU, and 0.944 for DL. Additionally, AMOS 27.0 was used to test for direct and mediated effects.

## 4. Results

### 4.1. Assessment of the Model Fit

This study uses a questionnaire method and aims to eliminate potential interference that may arise from a common-method bias. Common-method biases often arise from the same data collection source, consistent measurement environment or context, and inherent characteristics of the project itself, which can lead to unnatural covariation between predictors and standard variables. In order to effectively reduce the potential influence of subjectivity factors on the common-method bias, this study deliberately adopted an anonymous survey method to reduce the impact of the social expectation effect on respondents’ responses, thus encouraging respondents to provide more truthful and unbiased feedback. However, there may still be some degree of common-method bias in the data. In this paper, a Harman’s univariate test method is used to combine all variables for exploratory factor analysis, and the proportion of the first principal component obtained without rotation was 23.063%, which was within the acceptable range, indicating that the common method did not affect this study.

In this paper, AMOS27.0 was used to analyze the five factors of I-type EC, D-type EC, PU, PEU, and DL as the basic model; the results are shown in Table 1, and the relevant indicators of the model are as follows: Chi/df is 2.158, RMSEA is 0.057, CFI is 0.046, IFI is 0.912, and TLI is 0.904. All indicators met the requirements, and the model fit well.

Table 2 shows the mean value, standard deviation (SD), and correlation coefficient for each variable. In this paper, it is found that all variables are positively significant, which preliminarily confirms the above hypothesis.

### 4.2. Direct-Effect Test

Table 3 shows the results of the direct effects in this paper. I-type EC (β = 0.500, S.E. = 0.058, *p* < 0.001) and D-type EC (β = 0.419, S.E.= 0.069, *p* < 0.001) have a significant impact on the PU of international students in China. I-type EC (β = 0.612, S.E. = 0.064, *p* < 0.001) and D-type EC (β = 0.442, S.E. = 0.074, *p* < 0.001) have a significant impact on the PEU of international students in China. PU (β = 0.508, S.E. = 0.059, *p* < 0.001) and PEU (β = 0.227, S.E. = 0.064, *p* < 0.01) have a significant impact on the DL of international students in China.

### 4.3. Mediator Effect Test

To test the mediation effect, this paper follows the method of [14] ([14]) and uses AMOS 27.0 for the bootstrap test. Table 4 shows the results of the mediation effect of this paper, and PU mediates the relationship between I-type EC (indirect effect = 0.254, S.E. = 0.054, 95% CI = [0.152, 0.362] excluding 0), D-type EC (indirect effect = 0.213, S.E. = 0.050, 95% CI = [0.119, 0.315] excluding 0), and DL of international students. The PEU mediated I-type EC (indirect effect = 0.139, S.E. = 0.054, 95% CI = [0.042, 0.256] excluding 0), D-type EC (indirect effect = 0.101, S.E. = 0.044, 95% CI = [0.028, 0.201] excluding 0), and DL of international students ([31]).

## 5. Discussion

This paper investigates the mechanism of interaction between EC, PU, PEU, and DL, and the results mainly verify the following hypotheses:

First, EC has a positive effect on the DL of international students in China; specifically, both I-type EC and D-type EC can promote the DL of international students; that is, hypotheses 1a and 1b are supported, which also supports the studies on the impact of EC on digital learning ([10]; [23]). I-type EC (β = 0.472, S.E. = 0.051, *p* < 0.001) > D-type EC (β = 0.393, S.E. = 0.058, *p* < 0.001) is also found.

Second, whether it is I-type EC or D-type EC, EC can have a positive effect on the DL of international students in China under the mediating effect of PU. Therefore, hypotheses 2a and 2b are all accepted. This paper finds the main path of the mediating effect of PU: I-type EC→PU→DL (β = 0.254, *p* < 0.001) > D-type EC→PU→DL (β = 0.213, *p* < 0.001), which also confirms, to some extent, the important role of PU in digital teaching innovation found in previous studies ([4]; [6]; [18]).

Finally, whether it is I-type EC or D-type EC, EC can have a positive impact on the DL of international students in China under the mediating effect of PEU, so hypotheses 3a and 3b are considered valid, and this paper finds the main path of the mediating effect of PEU: I-type EC→PEU→DL (β = 0.139, *p* < 0.01) > D-type EC→PEU→DL (β = 0.101, *p* < 0.01), which to some extent confirms the significant effect of PEU on students’ digital learning found in previous studies ([3]; [5]; [18]; [20]).

## 6. Conclusions

Digital literacy is an important product of China’s comprehensive entry into the era of digital economy, and more and more policies in China have introduced digital literacy for citizens ([25]; [26]). From the perspective of cognitive and behavioral sciences ([24]; [33]; [52]), this paper analyzes the main paths of the formation of DL of international students in China by using a questionnaire method and structural equation model and finds the following main finding: EC driven by interest and EC caused by deprivation can stimulate international students’ DL, and also discusses the mediating role of PU and PEU on the positive relationship between the two.

The findings of this paper provide actionable insights for educational practitioners and policy makers. In terms of educational management, perceptual factors such as PU and PEU are crucial to cultivating the DL of international students in China, and teachers and staff should expand the application scenarios of embedding digital technology in the education of international students in China from the perspective of students’ cognition and acceptance in the process of teaching tool innovation. At the societal level, the cognitive antecedents of curiosity have a profound impact on the formation of the DL of international students in China, and the government and enterprises should further develop high-quality and story-line digital learning resources and provide professional development solutions for teachers and staff to improve DL, so as to promote the process of using digital resources to transform digital literacy into digital competence for international students.

This paper has the following implications: First, this paper discusses the formation mechanism of international students’ DL in China, explores the potential processes of students with cross-cultural backgrounds in the Chinese context, and extends the understanding of existing research focusing on teachers’ DL. Secondly, this paper enriches and extends the Technology Acceptance Model from the perspectives of I-type EC and D-type EC, constructs a theoretical framework of cognition–perception–formation of DL for international students in China, and verifies the positive impact of EC on DL, and the results of this study are also helpful to expand the educational environment in other fields. Finally, the findings of this paper also have important guiding significance for practical practice, which reflects the development process of international students’ cognition of digital resources into DL and can provide an important basis for the actual effectiveness of digital education in China.

However, this paper also has some limitations: On the one hand, this paper mainly uses the questionnaire method to investigate the DL of international students in China, but the open-ended questions for the respondents are limited, and more interview research will be conducted in the future according to the topic of this research to find out the factors and models affecting the DL of international students in China from this structured interview. On the other hand, this paper mainly focuses on international students studying in China and mainly explores the formation mechanism of DL from the perspective of EC. Future research can also be conducted from the perspective of other factors (such as cognitive flexibility, self-efficacy, etc.), where cognitive flexibility is the information-processing level to explain students’ independent understanding and knowledge transfer, and self-efficacy is the behavior speculation and judgment at the self-expectation level, and their influence mechanism as personality variables needs to be further explored.

## Figures and Tables

**Figure 1 behavsci-15-00286-f001:**
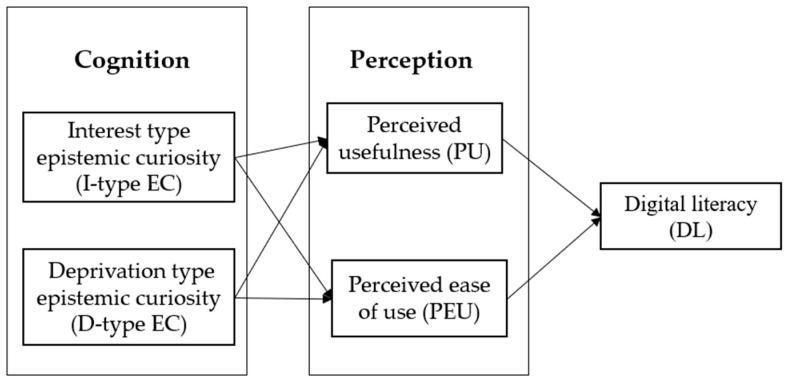
The framework of this paper.

**Table 1 behavsci-15-00286-t001:** Structural validity test.

Models	Factors	Chi/df	RMSEA	SRMR	CFI	IFI	TLI
Five-factor model	I-type EC, D-type EC, PU, PEU, DL	2.158	0.057	0.046	0.911	0.912	0.904
Four-factor model	I-type EC, D-type EC, PU + PEU, DL	2.545	0.066	0.053	0.881	0.882	0.872
Three-factor model	I-type EC + D-type EC, PU + PEU, DL	2.701	0.070	0.055	0.868	0.869	0.859
Two-factor model	I-type EC + D-type EC + PU + PEU, DL	3.146	0.078	0.059	0.833	0.834	0.823
Single-factor model	I-type EC + D-type EC + PU + PEU + DL	4.002	0.092	0.071	0.766	0.768	0.752

**Table 2 behavsci-15-00286-t002:** Correlation analysis.

	Mean	SD	1	2	3	4	5	6	7	8	9	10	11
Gender	1.538	0.499	1										
Age	2.867	1.035	−0.345 **	1									
Study stage	3.314	0.776	−0.233 **	0.689 **	1								
Major	1.725	0.654	−0.329 **	0.177 **	0.059	1							
Cumulative residence time in China	2.116	1.042	−0.104	0.154 **	0.060	−0.074	1						
Chinese study time	3.006	1.526	0.130 *	−0.381 **	−0.373 **	0.044	0.221 **	1					
I-type EC	3.721	0.860	−0.262 **	0.209 **	0.277 **	0.109 *	−0.034	−0.276 **	1				
D-type EC	3.567	0.739	−0.207 **	0.161 **	0.237 **	0.144 **	−0.016	−0.216 **	0.659 **	1			
PU	3.930	0.746	−0.113 *	0.157 **	0.267 **	0.046	−0.020	−0.165 **	0.609 **	0.591 **	1		
PEU	3.501	0.795	−0.210 **	0.138 **	0.223 **	0.122 *	−0.022	−0.222 **	0.677 **	0.603 **	0.614 **	1	
DL	3.811	0.634	−0.158 **	0.188 **	0.293 **	0.169 **	0.010	−0.151 **	0.620 **	0.590 **	0.720 **	0.631 **	1

Notes. * *p* < 0.05; ** *p* < 0.01.

**Table 3 behavsci-15-00286-t003:** Direct-effect results.

Hypothesis	Coefficients	S.E.	*p*-Value	Results
I-type EC→DL	0.472	0.051	***	Supported
D-type EC→DL	0.393	0.058	***	Supported
I-type EC→PU	0.500	0.058	***	Supported
I-type EC→PEU	0.612	0.064	***	Supported
PU→DL	0.508	0.059	***	Supported
D-type EC→PU	0.419	0.069	***	Supported
D-type EC→PEU	0.442	0.074	***	Supported
PEU→DL	0.227	0.064	**	Supported

Notes. ** *p* < 0.01, *** *p* < 0.001.

**Table 4 behavsci-15-00286-t004:** The mediation results.

Hypothesis	Coefficients	S.E.	Bias-Corrected 95%CI		*p*-Value	Results
			LL	UL		
I-type EC→PU→DL	0.254	0.054	0.152	0.362	***	Supported
I-type EC→PEU→DL	0.139	0.054	0.042	0.256	**	Supported
D-type EC→PU→DL	0.213	0.050	0.119	0.315	***	Supported
D-type EC→PEU→DL	0.101	0.044	0.028	0.201	**	Supported

Notes. ** *p* < 0.01, *** *p* < 0.001, LL = lower limit, UL = upper limit.

## Data Availability

The data presented in this study are available upon request from the corresponding author due to restrictions related to participant privacy concerns.

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
