# Peer review of "How Epistemic Curiosity Influences Digital Literacy: Evidence from International Students in China"

_behavsci, 2025, doi:10.3390/bs15030286_

Round 1
Reviewer 1 Report
Comments and Suggestions for Authors
Thank you for the opportunity to review this paper. I believe the strength of this paper can be found in its strong argument about the relationship between epistemic curiosity (EC), perceived usefulness (PU), perceived ease of use (PEU), and digital literacy (DL) among Chinese international students. The paper has contributed to the current scholarship with its use of a large sample size (353 international students) and appropriate statistical analyses (SPSS 27.0 and AMOS 27.0). While the paper makes a strong case for the role of EC in shaping DL, it acknowledges that other factors, such as cognitive flexibility and self-efficacy, may also influence digital literacy. To strengthen the methodological rigor of the study, I would suggest including a brief discussion of these additional factors, even if they were not directly measured. I would also clarify how the authors controlled for these and other potential variables.
I also have the impression that this paper could benefit from more concrete discussion of pedagogical suggestions. For instance, how might educators who work with Chinese international students use the findings in this study to develop more effective digital learning materials? Alongside the pedagogical suggestions, I also consider it beneficial to discuss potential cross-contextual applications. For example, the authors may discuss whether similar relationships between EC and DL might exist in other educational contexts or settings.
Finally, I suggest adding an appendix to share the complete questionnaire used in the study, which would be beneficial for readers who wish to gain a deeper understanding of the study's design.
Author Response
Thank you very much for your kind comments. We are also very thankful to your carefully reviewing the manuscript and stimulating comments. These constructive comments are very helpful for us to further improve this manuscript. Significant efforts were made to address necessary revisions in accordance with these comments. The major revisions on the figures and passages are shown in red in the new paper, while detailed explanations to the revisions are provided as below. We respond to the comments in “Point-to-Point Responses”.
Response to Reviewer #1.
Thank you for your valuable suggestions, to which we respond point by point as follows.
Comment No.1: While the paper makes a strong case for the role of EC in shaping DL, it acknowledges that other factors, such as cognitive flexibility and self-efficacy, may also influence digital literacy. To strengthen the methodological rigor of the study, I would suggest including a brief discussion of these additional factors, even if they were not directly measured. I would also clarify how the authors controlled for these and other potential variables.
Response to the reviewer’s comment No.1: Accepted.
Thank you for your highly constructive comments. We have greatly benefited from them and have followed your suggestions in revising the paper. As you said, we mentioned in the final conclusion section that cognitive flexibility and self-efficacy also have the potential to affect digital literacy. In this original manuscript, we expressed our expectations of this role in a short sentence, but such an expression does not conform to scientific norms. Based on your suggestions, we start from the concepts of cognitive flexibility and self-efficacy, and make theoretical derivations and future research plans in the conclusion part. We believe that such a manuscript ending can better conform to the scientific norms of academic research. For ease of understanding, we have highlighted the changes in red in the revised original. If this revision does not meet the needs, we look forward to your further guidance and continued refinement in the next round of revisions. Thanks again for the suggestion.
“However, this paper also has some limitations. On the one hand, this paper mainly uses the questionnaire method to investigate the DL of international students in China, but the open-ended questions for the respondents are limited, and more in-terview research will be conducted in the future according to the topic of this research to find out the factors and models affecting the DL of international students in China from this structured interview. On the other hand, this paper mainly focuses on inter-national students studying in China, and mainly explores the formation mechanism of DL from the perspective of EC. Future research can also be conducted from the per-spective of other factors (such as cognitive flexibility, self-efficacy, etc.), where cogni-tive flexibility is the information processing level to explain students’ independent understanding and knowledge transfer, and self-efficacy is the behavior speculation and judgement at the self-expectation level, and their influence mechanism as person-ality variables needs to be further explored.”
Comment No.2: I also have the impression that this paper could benefit from more concrete discussion of pedagogical suggestions. For instance, how might educators who work with Chinese international students use the findings in this study to develop more effective digital learning materials? Alongside the pedagogical suggestions, I also consider it beneficial to discuss potential cross-contextual applications. For example, the authors may discuss whether similar relationships between EC and DL might exist in other educational contexts or settings.
Response to the reviewer’s comment No. 2: Accepted.
Thank you for your highly constructive comments. We have greatly benefited from them and have followed your suggestions in revising the paper. We made revisions in two main ways. On the one hand, in the conclusion section,we add the actionable insights of this paper from the educational practitioners and societal levels, proposing the development of digital learning materials from the perspectives of EC, PU and PEU. On the other hand, we also mention that the discussion of EC and DL in this paper can be generalised to other educational settings. The specific changes are as follows:
“The findings of this paper provide actionable insights for educational practitioners and policy makers. In terms of educational management, perceptual factors such as PU and PEU are crucial to cultivating the DL of international students in China, and teachers and staff should expand the application scenarios of embedding digital tech-nology in the education of international students in China from the perspective of students’ cognition and acceptance in the process of teaching tool innovation. At the societal level, the cognitive antecedents of curiosity have a profound impact on the formation of DL of international students in China, and the government and enter-prises should further develop high-quality and story-line digital learning resources, and provide professional development solutions for teachers and staff to improve DL, so as to promote the process of using digital resources to transform digital literacy into digital competence for international students.”
“This paper has the following implications. First, this paper discusses the for-mation mechanism of international students’ DL in China, explores the potential pro-cesses of students with cross-cultural backgrounds in the Chinese context, and extends the understanding of existing research focusing on teachers’ DL. Secondly, this paper enriches and extends the Technology Acceptance Model from the perspectives of I-type EC and D-type EC, constructs a theoretical framework of cogni-tion-perception-formation of DL for international students in China, and verifies the positive impact of EC on DL, and the results of this study are also helpful to expand the educational environment in other fields. Finally, the findings of this paper also have important guiding significance for practical practice, which reflects the develop-ment process of international students’ cognition of digital resources into DL, and can provide an important basis for the actual effectiveness of digital education in China.”
Comment No.3: I suggest adding an appendix to share the complete questionnaire used in the study, which would be beneficial for readers who wish to gain a deeper understanding of the study's design.
Response to the reviewer’s comment No. 3: Accepted.
Thank you for noting this issue. We greatly appreciate the valuable suggestions of the reviewers and apologize for overlooking this issue. Given to your suggestion, we have added questionnaires for the main dimensions to facilitate a deeper understanding of the research design of this paper. The specific changes are as follows:
“Digital Literacy (Adapted from [38])
I can quickly, accurately, and comprehensively retrieve required course resources from the internet or databases.
I can effectively assess the reliability and quality of online information.
I can effectively organize and store collected course information for easy subse-quent use.
I can communicate with contacts using email, academic social networks (such as ResearchGate, LinkedIn), and other platforms.
I can collaborate across countries or institutions through online meetings and document collaboration tools.
I can participate in discussions and share experiences related to course content on blogs, discussion forums, or social media platforms.
I can express my opinions fluently and clearly during online communications.
I can use multimedia tools (such as video editing software, PowerPoint) to present my ideas in the form of original documents, audio, and video.
I can arrange complex course content and digitize it to effectively express my thoughts.
When using online resources, I consciously cite sources.
I do not disclose my own or others' information or sensitive data on the internet.
I can safely use course resources on different devices and networks (such as using VPNs, avoiding public wifi connections).
I can develop good internet habits to ensure physical and mental health and avoid internet addiction.
I understand internet ethics and related laws, and I adhere to these standards in my studies and work.
I can use various digital resources to enhance my course communication efficien-cy.
I have the ability to independently learn and master advanced digital tools to support my learning and work.
When needed, I can seek technical help online to solve digital technology issues encountered in my studies or work.
I can continuously improve my digital literacy as digital technology evolves.
Epistemic Curiosity(Adapted from [25-26,39])
I enjoy exploring digital technologies related to my courses.
I like learning digital tools or software that I am not familiar with.
I find new digital technologies and tools very attractive.
When learning new technologies, I like to delve into them and discover more ap-plications.
I enjoy discussing and exploring abstract concepts or cutting-edge theories in dig-ital technology.
When I encounter difficulties with new technologies in my courses, I must solve them before I can rest, even spending hours on it.
Abstract conceptual issues in my course’s digital technology make me keep thinking about how to answer them.
If I can’t solve digital technology issues in my courses, I feel frustrated, so I study even harder.
I act like a workaholic when it comes to digital technology problems that I believe must be solved.
I need to think for a long time to solve problems in digital technology.
Perceived Usefulness and Perceiving Ease of Use (Adapted from [42-43])
Finding and using digital technology resources effectively is very easy for me.
Learning content that interests me using digital technology is very easy for me.
I can relatively easily solve digital technology problems in my courses.
Using digital technology to improve my learning and work efficiency is very easy.
Using digital technology allows me to complete my work/study tasks more effi-ciently.
Using digital technology can make interactions with colleagues/classmates more efficient.
Using digital technology can enhance my initiative in learning or work.
Using digital technology can significantly improve my learning or work efficiency.”
Reviewer 2 Report
Comments and Suggestions for Authors
The study on how epistemic curiosity influences digital literacy among international students in China is highly relevant, addressing a critical aspect of education in an increasingly digital world. However, the paper could benefit from a clearer articulation of its unique contribution to the literature and a stronger theoretical framework linking epistemic curiosity and digital literacy. I have the following comments that need to be addressed:
1- The abstract needs improvements to include:
a. The main aim of the paper
b. The methodology
c. Brief findings
d. Contributions and originality.
e. The implications
2- The introduction needs improvements; this should be based on the research problem, questions, and theory discussions. The research problem and questions should be discussed in the introduction. Further, the research gap that justifies your contribution with theory underpinnings need to be discussed. It is necessary to discuss and mention how your research contributes to the strand literature.
3- The introduction in its current format lacks for supportive literature. There is very limited literature used in the introduction. The authors should include most recent and up-to-date research studies. The literature review in its current form is basic and simple. You should analyse, synthesis, and critically evaluate the research studies related to the topic to give a clear picture of the state of knowledge on the subject.
4-To enhance the visibility of the study's contributions, consider explicitly highlighting how the research advances the field of digital literacy in education (Al-Hattami, 2025) and its relevance to the specific context: international students in China.
5-The methodology needs more detail, particularly regarding sample selection, measurement tools, and potential biases.
6- The discussion section is missing. This section is very important, which compare the study results with past studies.
7- The article needs proofreading.
8- The conclusion needs to be revised. A discussion of the following points in several paragraphs should be provided:
a. Aim of the article
b. Methods applied and analysis tools
c. Major findings
d. Contribution, originality (implications)
f. Limitations
g. Future for future research
Reference
Al-Hattami, H. M. (2025). Understanding how digital accounting education fosters innovation: The moderating roles of technological self-efficacy and digital literacy. The International Journal of Management Education, 23(2), 101131.
Good luck
Comments on the Quality of English Language
Need improvements
Author Response
Thank you very much for your kind comments. We are also very thankful to your carefully reviewing the manuscript and stimulating comments. These constructive comments are very helpful for us to further improve this manuscript. Significant efforts were made to address necessary revisions in accordance with these comments. The major revisions on the figures and passages are shown in red in the new paper, while detailed explanations to the revisions are provided as below. We respond to the comments in “Point-to-Point Responses”.
Response to Reviewer #2.
Thank you for your valuable suggestions, to which we respond point by point as follows.
Comment No.1: The abstract needs improvements to include: a. The main aim of the paper; b. The methodology; c. Brief findings; d. Contributions and originality; e. The implications.
Response to the reviewer’s comment No.1: Accepted.
Thank you for your highly constructive comments. We have greatly benefited from them and have followed your suggestions in revising the paper. We have revised the abstract of this paper to include the above five parts you mentioned, as follows:
“Digital literacy is the core competitiveness and necessary ability that international students should cultivate while studying in China in the context of education digitalization, and this paper mainly explores whether epistemic curiosity can affect the digital literacy of international students in China. Based on the Technology Acceptance Model, this paper introduces the variable of ep-istemic curiosity, uses questionnaire survey method and quantitative tools (SPSS and AMOS software) to construct a model of the cognition-perception-formation mechanism of international students’ digital literacy in China, and obtains the following conclusions: firstly, both interest and deprivation type epistemic curiosity can directly promote the digital literacy of international students in China. Secondly, this paper discusses interest and deprivation type epistemic curi-osity can affect digital literacy under the mediating effect of perceived usefulness. Finally, per-ceived ease of use can also indirectly promote the relationship between epistemic curiosity and digital literacy of international students in China. The contribution of this paper is to highlight the formation mechanism of digital literacy in cross-cultural contexts, and to explore how interest and deprivation type epistemic curiosity affect the digital literacy of international students in China. To a certain extent, this paper reveals the potential process of international students in China to use digital resources to transform into digital literacy, and also provides useful evidence for the further development of attractive digital resources.”
Comment No.2: The introduction needs improvements; this should be based on the research problem, questions, and theory discussions. The research problem and questions should be discussed in the introduction. Further, the research gap that justifies your contribution with theory underpinnings need to be discussed. It is necessary to discuss and mention how your research contributes to the strand literature.
Response to the reviewer’s comment No.2: Accepted.
Thank you for noting this issue. We greatly appreciate the valuable suggestions of the reviewers and apologize for overlooking this issue. Given to your suggestion, we have made changes in two main ways: on the one hand, we have presented the research question in the introduction by adding questions; on the other hand, we have mentioned the research gaps in this paper in the marginal contributions section. The specific changes are as follows:
“Based on this, this study raises the following research questions: What factors influence the digital literacy of international students in China? How does epistemic curiosity affect the process mechanism of digital literacy formation of international stu-dents in China? Based on the Technology Acceptance Model, this study constructs a theoretical model of the digital literacy of international students in China with epis-temic curiosity to provide a useful reference for improving the learning effect and perceptual experience of international students in their home countries.
Since it is a relatively new area of research to promote digital literacy in education [14], and the marginal contributions of this study are as follows: On the one hand, this paper extends the antecedent variables of the Technology Acceptance Model and demonstrates the formation mechanism of international students' digital literacy in China from the perspective of epistemic curiosity (EC). On the other hand, based on the cross-cultural context and the perspective of international students, this paper discusses the relationship between perceived usefulness, perceived ease of use, inter-est type epistemic curiosity (I-type EC) and deprivation type epistemic curiosity (D-type EC) to explain the logical process of international students using digital re-sources to transform digital literacy.”
Comment No.3: The introduction in its current format lacks for supportive literature. There is very limited literature used in the introduction. The authors should include most recent and up-to-date research studies. The literature review in its current form is basic and simple. You should analyse, synthesis, and critically evaluate the research studies related to the topic to give a clear picture of the state of knowledge on the subject.
Response to the reviewer’s comment No.3: Accepted.
Thank you for your careful reading and helpful reminder. Based on your suggestion, we have made changes in two main areas: on the one hand, we have added references from the last three years, including the literature you provided Al-Hattami(2025); on the other hand, we have added a critical evaluation of the research on the topic, in order to get a clear picture of the state of knowledge on the topic. The details are as follows:
“In fact, digital literacy is a concept that has emerged with the development of technologies such as computer network media, digital twins, and human-computer collaborative teaching[14], and because its connotation is iteratively updated according to the global scientific and technological revolution and innovation, and changes in educational resources in different periods, scholars’ knowledge and understanding of it also be controversial [15-18], but they all agree on the important role of learners in collecting, processing and adding value to digital information in the digital age, which promotes the improvement of digital innovation capabilities in the field of educa-tion[19-21]. For this paper, digital literacy for international students in China refers to the digital learning competencies cultivated by international students during their second language learning in China, which is a comprehensive ability that integrates awareness, application, attitude, and behavior, and is fundamental and critical to the Sustainable Development Goals [5,22-23]. However, Unlike university students in their home countries, international students in China not only face academic pressure and digital competency tests when studying in their host countries, but also face cultural differences and digital adaptation challenges. Nami & Asadnia [24] found that cross-language learners interacting with AI bots can improve their language expres-sion and writing skills, and Wang et al. [13] found that international students with high digital literacy can improve their grades and academic performance while reduc-ing their perceived stress about technology.
Although there is a growing interest in the study of digital literacy in the context of second language learning, there is still little evidence on the mechanism of digital literacy formation of international students and less on the influencing factors of cog-nitive aspects (e.g. epistemic curiosity).”
Comment No.4: To enhance the visibility of the study's contributions, consider explicitly highlighting how the research advances the field of digital literacy in education (Al-Hattami, 2025) and its relevance to the specific context: international students in China.
Response to the reviewer’s comment No.4: Accepted.
Thank you for your highly constructive comments. We have greatly benefited from them and have followed your suggestions in revising the paper. Based on your suggestion, we have written clearly in the marginal contributions section about the study's contribution to the advancement of digital literacy in education, as well as the specific context of international students and cross-cultures, with the following modifications:
“Since it is a relatively new area of research to promote digital literacy in education [14], and the marginal contributions of this study are as follows: On the one hand, this paper extends the antecedent variables of the Technology Acceptance Model and demonstrates the formation mechanism of international students' digital literacy in China from the perspective of epistemic curiosity (EC). On the other hand, based on the cross-cultural context and the perspective of international students, this paper discusses the relationship between perceived usefulness, perceived ease of use, inter-est type epistemic curiosity (I-type EC) and deprivation type epistemic curiosity (D-type EC) to explain the logical process of international students using digital re-sources to transform digital literacy.”
Comment No.5: The methodology needs more detail, particularly regarding sample selection, measurement tools, and potential biases.
Response to the reviewer’s comment No.5: Accepted.
Thank you for your careful reading and helpful reminder. We have added the following to the methods and results section: on the one hand, we have clearly written out the three categories of research samples and the two stages of source for obtaining the samples; On the other hand, we have added two ways to avoid common method bias in the results section (Section 4.1), which are modified as follows:
“The participants in this paper are mainly international students studying in China. At present, there are three categories of Chinese students studying in China: prepara-tory students, undergraduate students and graduate students, among which prepara-tory students mainly study Chinese language and Chinese culture, and undergraduate and graduate students mainly study professional knowledge, and they study and live together with Chinese students. The research samples of this paper were collected through online questionnaires, and the data were sampled from October to November 2024 by snowballing and peer-led method, of which the first stage was mainly distrib-uted to professional course teachers, international students’ education management advisors and free distribution in each university, and the second stage was mainly col-lected through dormitory management teachers and compatriots of international stu-dents in China to ensure the cross-cultural background of international students’ data. A total of 398 questionnaires were collected through the online platform of Question-naire Star, and a total of 353 valid data samples of international students were collected in this study after excluding invalid questionnaires such as too short completion time (less than 120 seconds), single option, and repeated completion of the same IP.”
“This study uses a questionnaire method and aims to eliminate potential interfer-ence that may arise from common method bias. Common method bias often arises from the same data collection source, consistent measurement environment or context, and inherent characteristics of the project itself, which can lead to unnatural covaria-tion between predictors and standard variables. In order to effectively reduce the po-tential influence of subjectivity factors on common method bias, this study deliberately adopted an anonymous survey method to reduce the impact of the social expectation effect on respondents’ responses, thus encouraging respondents to provide more truthful and unbiased feedback. However, there may still be some degree of common methodological bias in the data. In this paper, Harman’s univariate test method is used to combine all variables for exploratory factor analysis, and the proportion of the first principal component obtained without rotation was 23.063%, which was within the acceptable range, indicating that the common method did not affect this study.”
Comment No.6: The discussion section is missing. This section is very important, which compare the study results with past studies.
Response to the reviewer’s comment No.6: Accepted.
Thank you for your highly constructive comments. We have greatly benefited from them and have followed your suggestions in revising the paper. Based on your suggestion, we have added a discussion section that includes a comparison of results and a comparison of previous research results, to make the research structure of this paper more complete and the research content more substantial. The specific changes as follows:
“5. Discussion
This paper investigates the mechanism of interaction between EC, PU, PEU and DL, and the results mainly verify the following hypotheses:
First, EC has a positive effect on the DL of international students in China, specif-ically, both I-type EC and D-type EC can promote the DL of international students, that is, hypotheses 1a and 1b are supported, which also supports the studies on the impact of EC on digital learning [29-30]. I-type EC (β=0.472, S.E.=0.051, P<0.001) > D-type EC (β=0.393, S.E.=0.058, P<0.001) is also found.
Second, whether it is I-type EC or D-type EC, EC can have a positive effect on the DL of international students in China under the mediating effect of PU. Therefore, hypotheses 2a and 2b are all accepted, this paper finds the main path of the mediating effect of PU: I-type EC→PU→DL (β=0.254, P<0.001)>D-type EC→PU→DL(β=0.213, P<0.001), which also confirms, to some extent, the important role of PU in digital teaching innovation found in previous studies [14,20,48-49].
Finally, whether it is I-type EC or D-type EC, EC can have a positive impact on the DL of international students in China under the mediating effect of PEU, so hypothe-ses 3a and 3b are considered valid, and this paper finds the main path of the mediating effect of PEU: I-type EC→PEU→DL (β=0.139, P<0.01) > D-type EC→PEU→DL (β=0.101, P<0.01), which to some extent confirms the significant effect of PEU on students’ digital learning found in previous studies[19,21,48,50].”
Comment No.7: The article needs proofreading.
Response to the reviewer’s comment No.7: Accepted.
Thank you for noting this issue. We greatly appreciate the valuable suggestions of the reviewers and apologize for overlooking this issue. Following your suggestions, we have improved the thesis in the following two ways. On the one hand, we further checked the language of the thesis to ensure the accuracy and quality of the English expression of the thesis. On the other hand, we contacted a language editing organisation to correct the language of the thesis to make it more native. We’ll move on to the next round if the language still doesn't meet the change requirements. Thank you once again for your suggestions.
Comment No.8: The conclusion needs to be revised. A discussion of the following points in several paragraphs should be provided:a. Aim of the article; b. Methods applied and analysis tools; c. Major findings; d. Contribution, originality (implications); f. Limitations; g.Future for future research
Response to the reviewer’s comment No.8: Accepted.
Thank you for your highly constructive comments. We have greatly benefited from them and have followed your suggestions in revising the paper. We have revised the abstract of this paper to include the above five parts you mentioned, as follows:
“Digital literacy is an important product of China's comprehensive entry into the era of digital economy, and more and more policies in China have introduced digital literacy for citizens[51-52]. From the perspective of cognitive and behavioral scienc-es[53-55], this paper analyses the main paths of the formation of DL of international students in China by using questionnaire method and structural equation model, and finds the following main findings: EC driven by interest and EC caused by deprivation can stimulate international students’ DL, and also discusses the mediating role of PU and PEU on the positive relationship between the two.
The findings of this paper provide actionable insights for educational practitioners and policy makers. In terms of educational management, perceptual factors such as PU and PEU are crucial to cultivating the DL of international students in China, and teachers and staff should expand the application scenarios of embedding digital tech-nology in the education of international students in China from the perspective of students’ cognition and acceptance in the process of teaching tool innovation. At the societal level, the cognitive antecedents of curiosity have a profound impact on the formation of DL of international students in China, and the government and enter-prises should further develop high-quality and story-line digital learning resources, and provide professional development solutions for teachers and staff to improve DL, so as to promote the process of using digital resources to transform digital literacy into digital competence for international students.
This paper has the following implications. First, this paper discusses the for-mation mechanism of international students’ DL in China, explores the potential pro-cesses of students with cross-cultural backgrounds in the Chinese context, and extends the understanding of existing research focusing on teachers’ DL. Secondly, this paper enriches and extends the Technology Acceptance Model from the perspectives of I-type EC and D-type EC, constructs a theoretical framework of cogni-tion-perception-formation of DL for international students in China, and verifies the positive impact of EC on DL, and the results of this study are also helpful to expand the educational environment in other fields. Finally, the findings of this paper also have important guiding significance for practical practice, which reflects the develop-ment process of international students’ cognition of digital resources into DL, and can provide an important basis for the actual effectiveness of digital education in China.
However, this paper also has some limitations. On the one hand, this paper mainly uses the questionnaire method to investigate the DL of international students in China, but the open-ended questions for the respondents are limited, and more in-terview research will be conducted in the future according to the topic of this research to find out the factors and models affecting the DL of international students in China from this structured interview. On the other hand, this paper mainly focuses on inter-national students studying in China, and mainly explores the formation mechanism of DL from the perspective of EC. Future research can also be conducted from the per-spective of other factors (such as cognitive flexibility, self-efficacy, etc.), where cogni-tive flexibility is the information processing level to explain students’ independent understanding and knowledge transfer, and self-efficacy is the behavior speculation and judgement at the self-expectation level, and their influence mechanism as person-ality variables needs to be further explored.”
Round 2
Reviewer 2 Report
Comments and Suggestions for Authors
The authors did well in their revision; the manuscript is much better now. Congratulations!
Comments on the Quality of English LanguageThe quality of English is good, but minor revision is preferred.
Author Response
Response to Reviewer 2 Comments
How Epistemic Curiosity Influences Digital Literacy:
Evidence from International Students in China(behavsci-3392851R2)
Point-to-Point Responses
Thank you for your valuable suggestions, to which we respond point by point as follows.
Comment No.1: The authors did well in their revision; the manuscript is much better now. Congratulations! The quality of English is good, but minor revision is preferred.
Response to the reviewer’s comment No.1: Accepted.
Thank you for your highly constructive comments. We have tried the language of the paper to meet the journal’s requirements for English language expression. Thank you again for your support.